# Dynamics of Circulating CD14/CD16 Monocyte Subsets in Obstructive Sleep Apnea Syndrome Patients upon Hypoglossal Nerve Stimulation

**DOI:** 10.3390/biomedicines10081925

**Published:** 2022-08-09

**Authors:** Ralph Pries, Christian Lange, Nicole Behn, Karl-Ludwig Bruchhage, Armin Steffen

**Affiliations:** Department of Otorhinolaryngology, University of Luebeck, 23538 Luebeck, Germany

**Keywords:** obstructive sleep apnea syndrome, CPAP therapy, HNS therapy, monocyte subsets

## Abstract

*Background:* Obstructive sleep apnea syndrome (OSAS) is a widespread respiratory disease that is associated with recurrent breathing intermissions at night. The corresponding oxidative stress triggers a low-grade systemic inflammation which leads to alterations of different immune cells in the peripheral blood. The current standard treatment for OSAS is continuous positive airway pressure (CPAP), whereas hypoglossal nerve stimulation (HNS) has been established as a second-line treatment option for CPAP failure. The aim of the study was to investigate the influence of HNS for OSAS patients on the distribution and differentiation of circulating monocyte subsets in connection with the clinical parameters. *Materials and Methods:* Therefore, a detailed analysis of the distribution of CD14/CD16 characterized monocyte subsets in the peripheral blood of OSAS patients before and after HNS therapy was performed by flow cytometry. Furthermore, values of BMI (body mass index), ODI (oxygen desaturation index), and ESS (Epworth Sleepiness Scale) were measured. *Results:* These OSAS patients significantly improved AHI and ESS scores under HNS. In addition, HNS revealed the potential to ensure normal distributions of blood monocyte subsets and even improved the monocyte dynamics in selected OSAS patients, but there were no significant correlations with AHI, ODI, HNS usage, and daytime sleepiness. *Conclusions:* We conclude that HNS-related positive effects on the oxygenation of the peripheral blood as well as affect the distribution of circulating monocyte subsets, but clinical OSAS correlations are missing. Far more individual clinical, cellular and molecular factors are involved in this sensitive and complex regulatory network and have to be elucidated in further studies.

## 1. Introduction

Obstructive sleep apnea syndrome (OSAS) is a substantial and highly prevalent sleep disorder and is associated with recurrent breathing intermissions. OSAS is characterized by a nocturnal reduction (hypopnea) or total stoppage (apnea) of the airflow in the upper airways and subsequent reoxygenation [1]. In addition to reduced quality of life, OSAS is a major risk factor for cardiovascular diseases, metabolic disorders, and cognitive dysfunctions [2]. Today’s standard treatment for OSAS is noninvasive continuous positive airway pressure (CPAP), which secures respiratory support. Usage time of 4 h per night is considered sufficient for successful therapy in terms of daytime sleepiness and the risk for cardiovascular disease [3,4,5,6]. Hypoglossal nerve stimulation (HNS) in patients with OSA and CPAP failure has been applied since the 1990s, mainly the STAR trial of 2014 [7] led to the increased clinical application and intensive scientific interest [8]. The apnea–hypopnea index (AHI) between 15 and 65/h as well as the exclusion of a complete concentric collapse in a drug-induced sleep endoscopy have to be highlighted as inclusion criteria. Most of the studies have been carried out in patients with an overweight limited to a body mass index (BMI) of 35 kg/m^2^.

OSAS-associated intermitted hypoxia and oxidative stress are known to induce low-grade systemic inflammation and trigger increased activity of different types of immune cells, for example, lymphocytes and monocytes, and increased levels of pro-inflammatory cytokines and surface adhesion molecules [9,10,11,12,13,14]. OSAS-induced hypoxic stress increases the invasion of monocytes into the endothelium, the transformation of macrophages into foam cells, and the emergence of atherosclerosis and cardiovascular disease [15,16,17]. Recently, we have shown strongly decreased percentages of CD14^++^CD16^−^ classical monocytes in OSAS patients in correlation with increased abundances of both CD16^+^ monocyte subsets and a disturbed PD-1/PD-L1 crosstalk with peripheral blood T cells compared to healthy individuals [18].

Monocytes come from bone marrow hematopoietic stem cells and are part of the mononuclear phagocyte system [19]. They can be subdivided in terms of the subset-specific CD14 and CD16 surface expression [20,21,22]. CD14^++^CD16^−^ “classical” monocytes are classified as “naïve-like” monocytes. The CD14^+^CD16^+^ “intermediate” and CD14^dim+^CD16^+^ “non-classical” monocyte subsets are more differentiated and pro-inflammatory monocytes that reveal specific immune functions such as viral defense and patrolling [23,24]. In healthy individuals, they both account for about 5–10% of peripheral blood monocytes. Increased abundances of pro-inflammatory CD16^+^ subsets have been associated with different acute and chronic inflammatory diseases [25,26]. The three circulating monocyte subsets all have the ability to acquire macrophage morphology and characteristics, but the exact potential of each subset is still unclear [23]. Further investigations revealed a restoration of the CD14/CD16 monocyte subset abundances to normal levels as well as a decrease in monocytic PD-L1 expression in response to continuous positive airway pressure (CPAP) therapy [25]. However, the dynamics of monocyte subsets in the peripheral blood of OSAS patients upon HNS as a second-line therapy have not been investigated so far. The aim of the study was to analyze the influence of the opening of the upper airway by HNS in OSAS patients on the differentiation patterns of the peripheral blood monocyte subsets. Immunological data were compared with OSAS-associated clinical parameters and quality of life. The study aimed to better understand the immunological consequences of oxidative stress and the impact of the available clinical treatment options.

## 2. Materials and Methods

### 2.1. Ethics Statement

Patients with OSAS were selected, implanted, and examined during follow-up at the Department of Otorhinolaryngology, University Hospital Schleswig-Holstein, Campus Luebeck, after their written informed consent. The study was approved by the ethics committee of the University of Luebeck (approval number 21-183) and performed according to the ethical principles of the WMA Declaration of Helsinki.

### 2.2. Blood Collection and Clinical Data

Blood samples were obtained from OSAS patients before HNS implantation and after 6 to 12 months after surgical treatment. BMI (body mass index), AHI (apnea–hypopnea index), ODI (oxygen desaturation index), and HNS usage time were recorded before and after HNS therapy. Additionally, daytime sleepiness of OSAS patients was evaluated before and after HNS treatment using the established Epworth Sleepiness Scale (ESS) questionnaire [27], which measures the probability of falling asleep in a variety of situations and indicates underlying sleep disorders or medical conditions [28].

### 2.3. FACS Analysis of Monocyte Subsets in Whole Blood

Peripheral blood samples were collected by venipuncture into sodium citrate containing S-Monovette (Sarstedt; Nümbrecht, Germany). Within 4 h after blood collection, 20 µL of citrate blood was diluted in 80 µL phosphate-buffered saline (PBS). Blood cells were stained with the listed antibodies: CD45-PE, CD14-FITC, CD16-BV-510, HLA-DR-APC-Cy7, and CD3-PerCP (all from Biolegend, San Diego, CA, USA). After 25 min staining in the dark, 650 µL RBC Lysis Buffer (Biolegend) was added, and samples were further incubated for 20 min. Afterward, the samples were centrifuged at 400× *g* for 5 min and supernatant was discarded. Next, cell sediments were resuspended in 100 µL fresh PBS and analyzed by flow cytometry. Flow cytometry was performed using a MACSQuant 10 flow cytometer (Miltenyi Biotec, Bergisch-Gladbach, Germany) and raw data were analyzed with FlowJo^TM^ software version 10.0 (FlowJo, LLC, Ashland, USA).

### 2.4. Statistical Analysis

Statistical analyses were conducted using GraphPad Prism Version 7.0f (GraphPad Software, Inc., San Diego, CA, USA) and paired Student’s *t*-tests for pairwise comparison of data before and after HNS treatment. The correlation between different clinical or cellular parameters was calculated using multivariate regression with Pearson correlation coefficient. *p* < 0.05 (*), *p* < 0.01 (**), and *p* < 0.001 (***).

## 3. Results

### 3.1. Patients Characteristics and Response to HNS Therapy

In this study, we analyzed the abundance of blood monocyte subsets of 16 OSAS patients (6 female/10 male) with a median age of 56.8 years. Blood samples were collected from OSAS patients before HNS surgical treatment as well as after 6 to 12 months of HNS therapy. BMI (body mass index), AHI (apnea–hypopnea index), ODI (oxygen desaturation index), and HNS usage time were measured before and after HNS therapy.

Initial ESS values showed a mean of 13.9 points, whereas values below 11 are considered normal. About 44% of the patients suffered from a moderate (AHI 15–30/h) and 56% presented a severe OSAS (AHI ≥ 30/h). According to WHO guidelines two patients displayed a normal weight (BMI 20 to 25 kg/m^2^), whereas seven patients were overweight (BMI 26 to 30 kg/m^2^), and seven had adiposity I° (BMI 31to 35 kg/m^2^).

The analyzed patient cohort revealed overall significantly improved OSAS severity in AHI and ODI as well as daytime sleepiness using ESS values upon HNS treatment, respectively (Figure 1).

In addition, the average HNS usage time (h/week) was assessed by telemetry read-out from the implant. Data revealed an overall average use of 45.8 h per week (range 13 h to 67 h), whereas 13 of 16 examined patients used HNS for at least 4 h per night which is considered sufficient for successful therapy with regard to daytime sleepiness and quality of life. Three patients revealed an HNS usage time of less than 4 h per night. Further correlation analysis revealed significant correlations between HNS usage time (h/week) and measured AHI and ODI values, respectively (Figure 2).

### 3.2. Monocyte Subset Distribution upon HNS Therapy

We analyzed the abundances of CD14^++^CD16^−^ (classical), CD14^++^CD16^+^ (intermediate) and CD14^dim+^CD16^+^ (non-classical) monocyte subsets prior and 6 to 12 month after surgical HNS treatment. The gating strategy of CD14- and CD16-defined monocyte subsets was performed as published before [18]. All analyzed patients were pretreated with CPAP respiratory support before changing to HNS treatment because of different reasons (Appendix A Appendix A).

Thus, most patients (n = 10) revealed a normal distribution of blood monocyte subsets prior to HNS treatment. Six patients showed a severe drop in classical monocytes accompanied by an increase in intermediate and non-classical monocytes prior to HNS treatment. After 6 to 12 months of HNS treatment flow cytometric measurements revealed a reduced number of patients (n = 3) with monocyte subset alterations (Figure 3).

Additional individual patient-related investigations identified a significant reconstitution of the monocyte subset abundances of five OSAS patients (O75, O80, O99, O103, O105) and unchanged levels of one patient (O71) upon HNS therapy for at least 6 months. On the other hand, flow cytometric analysis revealed a significant drop in classical monocytes accompanied by increased abundances of intermediate and non-classical subsets in two OSAS patients (O93, O98) after HNS therapy (Figure 4).

Next, correlation analysis between the different initial clinical parameters measured in OSAS patients prior to HNS treatment and the identified monocyte subsets distribution was performed with the Pearson correlation coefficient. Data revealed no significant overall correlations between monocyte subset abundances and BMI, AHI, ODI, or ESS, respectively (Figure 5).

In summary, our data indicate that CPAP pretreated OSAS patients who did not tolerate this therapeutic option, revealed significantly improved AHI and ESS values upon HNS treatment. Furthermore, HNS treatment has the potential to ensure normal distributions of blood monocyte subsets and even improve the monocyte dynamics in certain OSAS patients. On the other hand, two patients revealed a drop in classical monocyte percentages, without any significant correlation with OSAS-related clinical parameters.

## 4. Discussion

The aim of the present study was to investigate the influence of HNS therapy on percentages of monocyte subsets in the peripheral blood of OSAS patients using flow cytometry. In a recent study, we have shown that OSAS leads to a significant decrease in circulating classical monocytes associated with increased percentages of the CD16^+^ subsets [18]. An increase in pro-inflammatory CD16^+^ monocyte subsets has been linked to various inflammatory conditions [20], whereas increased percentages of intermediate monocytes comparable to OSAS patients were up to now only found in asthma patients [26]. OSAS-related oxidative stress triggers increased activities of pro-inflammatory transcription factors such as NFκB and HIF-1α [29]. The impact of respiratory support in OSAS patients has been so far solely evaluated in terms of atherosclerosis-associated inflammatory cytokines, since this is the most common secondary disease of OSAS patients [30]. Several studies revealed significantly increased TNF-α, CRP, IL-6, IL-8, VCAM, ICAM, and E-selectin levels in OSAS patients which decreased in response to at least 3 months of CPAP treatment [31,32,33].

It is well known that there is a coherence between obesity and obstructive sleep apnea syndrome (OSAS). OSAS gives rise to reduced physical activity and metabolic dysfunction and thus promotes obesity. The other way round, obesity is a major risk factor for OSAS [34]. The synergistic effect of obesity and OSAS increased the incidence of metabolic diseases, such as dyslipidemia, hypertension, insulin resistance, cardiovascular diseases, and non-alcoholic fatty liver disease [35]. These observations corroborate data from Ng and colleagues which identified decreased expressions of adiponectin and irisin levels in obese OSAS patients in response to CPAP treatment for 3 months [36]. In our study here, we had a typical BMI distribution in an OSAS cohort treated with HNS [37,38]. Furthermore, about 80% of children with Marfan’s syndrome suffer from OSAS, which is characterized by a present reduction in palatal area and volume and thus underlines the multifactorial etiology of obstructive sleep apnea syndrome [39].

We have recently shown, that OSAS patients revealed a significant restoration of monocyte subsets and monocytic PD-L1 expression upon CPAP treatment [25]. The present study is the first to address OSAS-induced immunological changes in terms of the influence of HNS therapy on a cellular level.

We found a normal distribution of blood monocyte subsets in the majority of CPAP pretreated OSAS patients which could be successfully obtained over the analyzed period of 6 to 12 months of HNS treatment. Here, it is unclear what time period is needed for diminishing the CPAP effect, even in insufficient but minimal usage before HNS implantation. Further, five patients revealed decreased levels of classical monocytes prior to HNS treatment and a significant improvement after 6 to 12 months of therapy. These findings are most likely associated with better oxygenation at night and reduced oxidative stress as seen by the improved AHI. On the other hand, one patient showed no effects and two patients revealed worsened values of monocyte subset abundances, although these patients also revealed improved AHI and ODI values. Overall correlation analysis did not reveal significant correlations between monocyte distribution and the different clinical parameters measured in OSAS patients. These data suggest that in addition to the investigated clinical parameters BMI, AHI, or ODI, many other clinical, as well as individual molecular factors and comorbidities such as atherosclerosis, diabetes, fatty liver or chronic inflammations, must be taken into account in further future investigations in larger patient cohorts. Additionally, the normalization of the monocyte patterns which we found in PAP therapy [25] needs a stricter normalization than provided by HNS despite clinical improvements. It remains unclear what residual OSAS is tolerated for normal monocyte distribution patterns. As PAP opens the entire upper airway with additional thoracic effects, HNS is only effective to treat the upper airway collapse. Therefore, adipositas-related hypoventilation is insufficiently addressed with HNS, especially in an overweight cohort like ours here.

A limitation of our study is the relatively small patient cohort analyzed. Further comprehensive investigations on larger cohorts of OSAS patients are required to better understand the interplay of circulating immune cells and the different clinical parameters of HNS therapy. A larger patient cohort could also identify further correlations between monocyte subset abundances and clinical parameters such as BMU, AHI, or ODI. Our data underline that respiratory support of OSAS patients using HNS is able to counteract and stabilize the oxidative stress-related imbalance on circulating monocytes whereas the responsible molecular mechanisms remain elusive. We assume that improved oxygenation of the peripheral blood in response to HNS treatment as well improves the distribution of circulating monocyte subsets, but that a holistic view of the clinical, cellular, and molecular individual parameters is required to understand the mechanisms of OSAS-related immune alterations and their treatment.

## Figures and Tables

**Figure 1 biomedicines-10-01925-f001:**
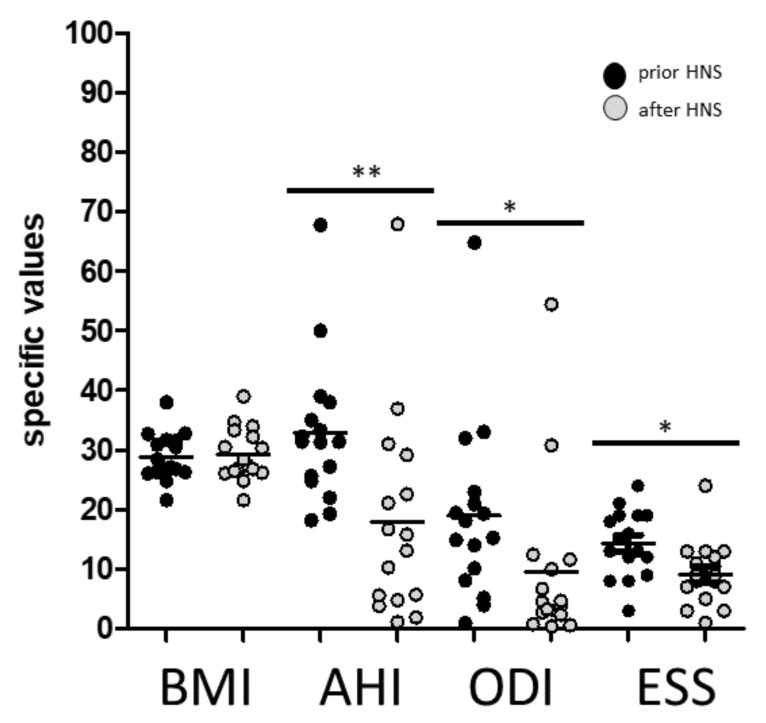
Values of BMI (body mass index), AHI (apnea–hypopnea index), ODI (oxygen desaturation index), and Epworth Sleepiness Scale (ESS) before and after HNS therapy. *: *p* < 0.05; **: *p* < 0.01. Post hoc power calculation was performed for the significant differences and revealed values of 100% for both AHI and ODI.

**Figure 2 biomedicines-10-01925-f002:**
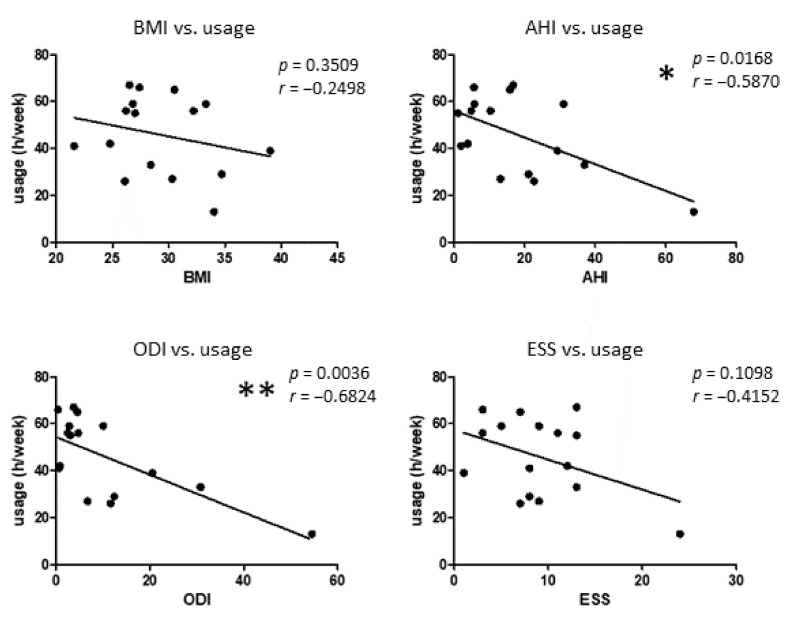
Correlation analysis between HNS usage time (h/week) and BMI (body mass index), AHI (apnea–hypopnea index), ODI (oxygen desaturation index), and Epworth Sleepiness Scale (ESS) before and after HNS therapy, respectively. A multivariate progression with the Pearson correlation was performed. The correlation coefficient (r) and *p*-values are shown for each correlation. *: *p* < 0.05; **: *p* < 0.01.

**Figure 3 biomedicines-10-01925-f003:**
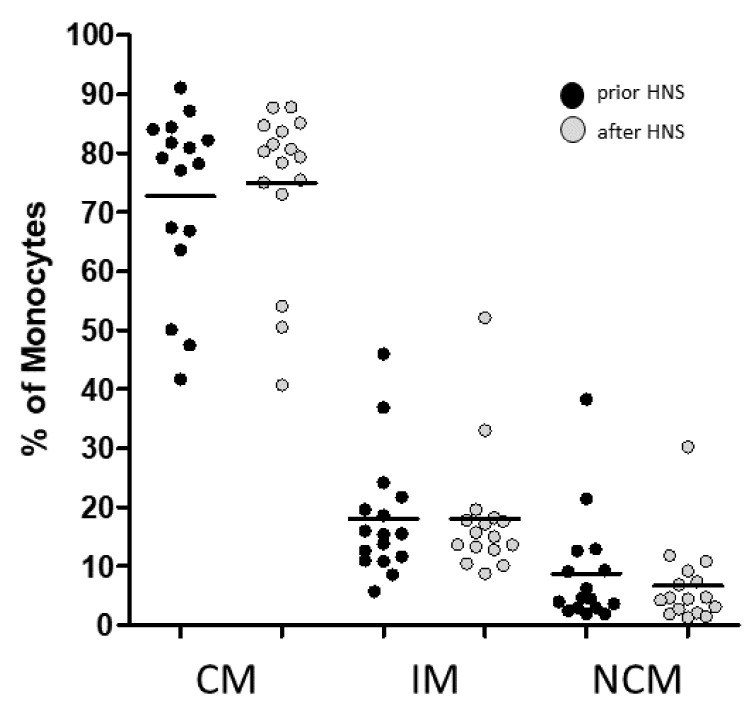
Flow cytometric analysis of monocyte subsets. Monocyte measurements from whole blood samples revealed the percentages of classical monocytes (CM), CD16^+^ intermediate (IM), and non-classical monocytes (NCM) in OSAS patients prior to and after HNS treatment.

**Figure 4 biomedicines-10-01925-f004:**
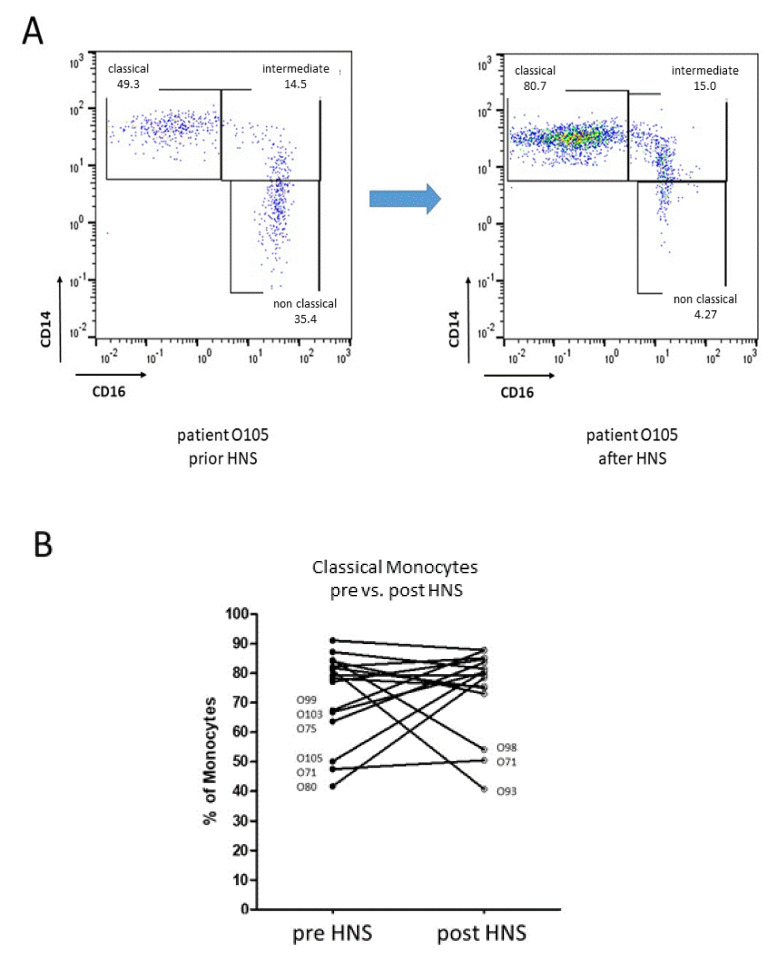
Flow cytometric analysis of classical monocytes before (pre) and after (post) HNS treatment. (**A**): Example flow cytometry gating scheme of peripheral monocyte subsets of patient O105 before and after 12-month HNS treatment. (**B**): Whole blood analysis revealed an improvement in five patients, a decrease in classical monocytes in two patients, and an unchanged level in one patient upon HNS treatment. Patients’ numbers are given in case of redistributed monocyte abundances.

**Figure 5 biomedicines-10-01925-f005:**
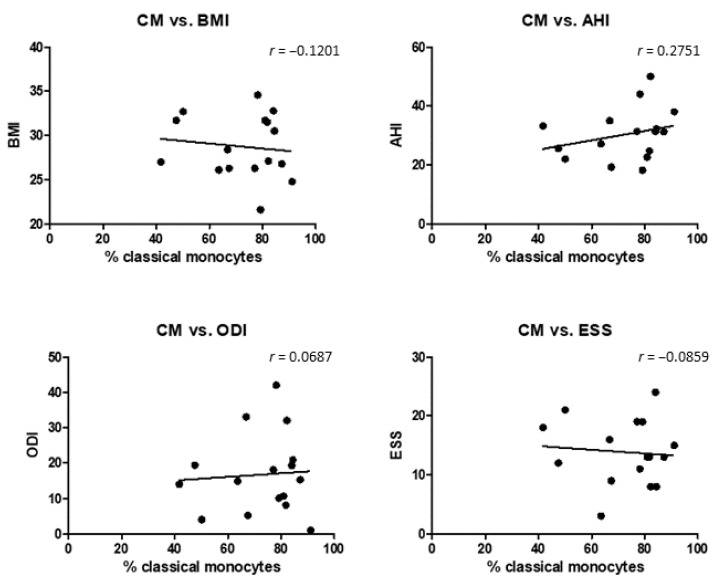
Correlation analysis between different parameters (BMI, AHI, ODI, ESS) measured in OSAS patients. A multivariate progression with the Pearson correlation was carried out. The correlation coefficient (r) is given for each pair.

## Data Availability

The data presented in this study are available on request from the corresponding author.

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
