# Peer review of "Dynamics of Circulating CD14/CD16 Monocyte Subsets in Obstructive Sleep Apnea Syndrome Patients upon Hypoglossal Nerve Stimulation"

_biomedicines, 2022, doi:10.3390/biomedicines10081925_

Round 1

Reviewer 1 Report

Dear Authors 

The aspects of this paper are very specific and very well written. The paper is interesting and it needs some minor revisions:

- Introduction : line 41-47 the same sentence is repeated with two different references, correct and clarify the matter

- Reference : the numbering is wrong and some articles are too old. 

Howewer the article strongly recommended reference is : Obstructive sleep apnea in children with Marfan syndrome: Relationships between three-dimensional palatal morphology and apnea-hypopnea index
Paoloni V, Cretella Lombardo E, Placidi F, Ruvolo G, Cozza P, Laganà G. International Journal of Pediatric OtorhinolaryngologyVolume 112, Pages 6 - 9, September 2018

Author Response

Dear Authors 

The aspects of this paper are very specific and very well written. The paper is interesting and it needs some minor revisions:

            Dear Reviewer, thank you for you constructive and positive comments.

- Introduction : line 41-47 the same sentence is repeated with two different references, correct and clarify the matter

Dear Reviewer, we are sorry for this repetition, which occurred somehow during the final manuscript formation. Has been corrected.

- Reference : the numbering is wrong and some articles are too old. 

References and their numbering have been carefully revised.

Howewer the article strongly recommended reference is : Obstructive sleep apnea in children with Marfan syndrome: Relationships between three-dimensional palatal morphology and apnea-hypopnea index
Paoloni V, Cretella Lombardo E, Placidi F, Ruvolo G, Cozza P, Laganà G. International Journal of Pediatric OtorhinolaryngologyVolume 112, Pages 6 - 9, September 2018

            The reference has been added to the manuscript as suggested.

Reviewer 2 Report

This is an interesting study assessing the dynamics of circulating CD14/CD16 monocytes in people with obstructive sleep apnea syndrome. This manuscript is well written. The following comments are provided for the authors to improve this manuscript. 

Major: Have the authors performed post hoc power analysis? What is the power achieved for the significant differences noted from a sample size of 16? Please incorporate power analysis in the manuscript.

Data revealed no significant overall correlations between monocyte subset abundances and BMI, AHI, ODI or ESS, respectively (Figure 5). - I agree with this conclusion, but does it perhaps because of the small sample size? This should be discussed.

Minor:

Usage time of 4 hours per night has been shown the minimum to reduce daytime sleepiness and the risk for cardiovascular disease. - Please consider rephrase this sentence.

Can the authors provide the raw FACS data as supplementary information?

Author Response

This is an interesting study assessing the dynamics of circulating CD14/CD16 monocytes in people with obstructive sleep apnea syndrome. This manuscript is well written. The following comments are provided for the authors to improve this manuscript. 

            Dear Reviewer, thank you for you constructive and positive comments.

Major: Have the authors performed post hoc power analysis? What is the power achieved for the significant differences noted from a sample size of 16? Please incorporate power analysis in the manuscript.

Post hoc power analysis has been added to the significant differences in Fig. 1 as suggested.

Data revealed no significant overall correlations between monocyte subset abundances and BMI, AHI, ODI or ESS, respectively (Figure 5). - I agree with this conclusion, but does it perhaps because of the small sample size? This should be discussed.

The limitation of the study due to the small sample size has been added to the discussion as suggested.

Minor:

Usage time of 4 hours per night has been shown the minimum to reduce daytime sleepiness and the risk for cardiovascular disease. - Please consider rephrase this sentence.

The sentence has been rephrased as suggested.

Can the authors provide the raw FACS data as supplementary information?

Raw FACS data have been added as supplementary information to illustrate the flow cytometric identification of the CD14/CD16 monocyte subsets.